# UNSUPERVISED LEARNING OF SENTENCE REPRESENTATIONS USING SEQUENCE CONSISTENCY

## ABSTRACT

Computing universal distributed representations of sentences is a fundamental task in natural language processing. We propose ConsSent, a simple yet surprisingly powerful unsupervised method to learn such representations by enforcing consistency constraints on sequences of tokens. We consider two classes of such constraints – sequences that form a sentence and between two sequences that form a sentence when merged. We learn sentence encoders by training them to distinguish between consistent and inconsistent examples, the latter being generated by randomly perturbing consistent examples in six different ways. Extensive evaluation on several transfer learning and linguistic probing tasks shows improved performance over strong unsupervised and supervised baselines, substantially surpassing them in several cases. Our best results are achieved by training sentence encoders in a multitask setting and by an ensemble of encoders trained on the individual tasks.

## 1 INTRODUCTION

In natural language processing, the use of distributed representations has become standard through the effective use of word embeddings. In a wide range of NLP tasks, it is beneficial to initialize the word embeddings with ones learned from large text corpora like word2vec (Mikolov et al. (2013)) or GLoVe (Pennington et al. (2014)) and tune them as a part of a target task e.g. text classification. It is therefore a natural question to ask whether such standardized representations of whole sentences that can be widely used in downstream tasks is possible.

There are several of approaches to this problem. Taking cue from word2vec, an unsupervised learning approach is taken by SkipThought (Kiros et al. (2015)), FastSent (Hill et al. (2016)) and Quick-Thoughts (Logeswaran & Lee (2018)), exploiting the closeness of adjacent sentences in a text corpus. Models trained using a language modeling objective have also been shown to produce excellent sentence representations (Ruder & Howard (2018); Radford (2018)). More recently, the work of Conneau et al. (2017) takes a supervised learning approach. They train a sentence encoder on large scale natural language inference datasets (Bowman et al. (2015); Williams et al. (2018)) and show that the learned encoding perform well on a set of transfer tasks. This is reminiscent of the approach taken by ImageNet (Deng et al. (2009)) in the computer vision community. In Subramanian et al. (2018), the authors train a sentence encoder on multiple tasks to get improved performance.

In this paper, we take a slightly different unsupervised approach to learning sentence representations. We define a sequence of tokens to be *consistent* if they form a valid sentence. While defining consistency precisely is difficult, we generate approximately inconsistent sequences of tokens by randomly perturbing consistent ones. We extend the notion of consistency to pairs of sequences – two sequences are defined to be consistent if they can me merged to form a consistent sequence. We then train a sentence encoder to discriminate between consistent and inconsistent sequences or pairs of sequences. Note that external supervised labels are not required for training as we generate our own labels using the notion of consistency. We call our proposed method *ConsSent*.

We generate inconsistent sequences in four different ways – by deleting, permuting, inserting and replacing random tokens in a consistent sequence. For pairs of sequences, we generate inconsistent examples by randomly partitioning sentences into two parts and pairing parts from two different sentences, where the method of partitioning can be contiguous or non-contiguous. We vary the

complexity of the training tasks by choosing a different number of tokens and pairs to generate inconsistent examples and train several sentence encoders on a set of unordered sentences from the Billionword corpus (Chelba et al. (2014)). We also combine the training tasks and encoders in two different ways – one by training an encoder in a multitask setting and the other by creating an ensemble of models trained on the individual tasks.

We evaluate the trained sentence encoders on a set of transfer learning and linguistic probing tasks encapsulated in the SentEval benchmark (Conneau & Kiela (2018)). Our models achieve strong results in both settings, improving upon existing unsupervised and supervised methods of learning sentence representations in several cases.

## 2  RELATED WORK

Learning general sentence encoders is a fundamental problem in NLP and there is a long list of works that addresses this problem. To begin with, an untrained BiLSTM model with max pooling of the intermediate states performs fairly well on several transfer and linguistic probing tasks (Conneau et al. (2018)). The next set of simple models consists of a bag-of-words approach using learned word embeddings like GloVe (Pennington et al. (2014)) or FastText (Joulin et al. (2017)), where a simple average of the word embeddings in a sentence define the sentence embedding. Although very fast, these approaches are limited by the dimensions of the word embeddings and do not take into account the order of the words.

Due to the availability of practically unlimited textual data, learning sentence encoders using unsupervised learning is an attractive proposition. The SkipThought model of Kiros et al. (2015) learns sentence encoders by using an encoder-decoder architecture. Exploiting the relatedness inherent in adjacent sentences, the model is trained by using the encoder to encode a particular sentence and then using the decoder to decode words in adjacent sentences. This approach is directly inspired by a similar objective for learning word embeddings like word2vec (Mikolov et al. (2013)). In more recent work, Logeswaran & Lee (2018) propose QuickThoughts, which uses a form of discriminative training on encodings of sentences, by biasing the encodings of adjacent sentences to be closer to each other than non-adjacent sentences, where closeness is defined by the dot product. The Bag-of-Words approach is developed further in the FastSent model (Hill et al. (2016)) which uses a bag-of-words representation of a sentence to predict words in adjacent sentences. In work by Arora et al. (2017) it is shown that a simple post processing of the average word-embeddings can perform comparably or better than skip-thought like objectives. The work of Pagliardini et al. (2018), where the authors use n-gram features for unsupervised learning is also relevant. The other notable unsupervised approach is that of using a denoising autoencoder (Hill et al. (2016)).

Language modeling is a very natural task for natural language for which models can be trained in an unsupervised manner. The context encoder trained on a large corpus with a language modeling objective can be used to generate sentence representations by encoding the sentence as a context. Work by Ruder & Howard (2018) and Radford (2018) has shown that training language models on large text corpora produce excellent sentence encoders that perform very well on downstream tasks. The recent work of Devlin et al. (2018) pushes this approach even further by training large transformers (Vaswani et al. (2017)) on very large text corpora using a slightly modified masked language modeling objective. The language modeling based approaches typically require corpora with ordered sentences or document level corpora, which is not a requirement in our case.

Supervised learning has been used to learn better sentence encoders. Conneau et al. (2017) use the SNLI (Bowman et al. (2015)) and MultiNLI (Williams et al. (2018)) corpus to train a sentence encoder on the natural language inference task that performs well on several transfer tasks. Another domain where large datasets for supervised training is available is machine translation and the work of McCann et al. (2017) exploits this in learning sentence encoders by training it on the machine translation task. Subramanian et al. (2018) combine several supervised and unsupervised objectives in a multi-task framework to obtain some of the best results on learning general sentence representations.

Our approach is based on automatically generating (possibly noisy) training data by perturbing sentences. Such an approach was used by Wagner et al. (2009) to train a classifier to judge the grammaticality of sentences. The ungrammatical sentences were generated by, among other things, dropping

and inserting words. Recent work by Warstadt et al. (2018) extend this approach by using neural network classifiers. Finally, in parallel to our work, a recent report Ranjan et al. (2018) uses word dropping and word permutation to generate fake sentences and learn sentence encoders. Our work is substantially more general and exhaustive.

## 3    CONSISTENT SEQUENCES AND DISCRIMINATIVE TRAINING

Let $\mathbf{S} = \{w_1, w_2, \cdots, w_n\}$ be an ordered sequence of $n$ tokens. We define this sequence to be *consistent* if the tokens form a valid sentence. Let $\mathbf{E}$ be an encoder that encodes any sequence of tokens into a fixed length distributed representation. Our goal is to train $\mathbf{E}$ to discriminate between consistent sequences and inconsistent ones. We argue that such an encoder will have to encapsulate many structural properties of sentences, thereby resulting in a good sentence representation.

Whether a sequence of tokens $\mathbf{S}$ is consistent is a notoriously hard problem to solve. We take a slightly different approach. We start from a consistent sequence (e.g. from some text corpus) and introduce random perturbations to generate sequences $\mathbf{S}'$ that are not consistent. We take inspiration from the standard operations in string edit distance and consider the following variations for generating $\mathbf{S}'$.

- ConsSent-D($k$): We pick $k$ random tokens in $\mathbf{S}$ and *delete* them.
- ConsSent-P($k$): We pick $k$ random tokens in $\mathbf{S}$ and *permute* them randomly (avoiding the identity permutation).
- ConsSent-I($k$): We pick $k$ random tokens *not* from $\mathbf{S}$ and *insert* them at random positions in $\mathbf{S}$.
- ConsSent-R($k$): We pick $k$ random tokens in $\mathbf{S}$ and replace them with other random tokens not in $\mathbf{S}$.

It is important to note that it is possible that in some cases $\mathbf{S}'$ will itself form a valid sentence and hence violate the definition of inconsistency. We do not address this issue and assume that such cases will be a relatively few and will not influence the encoder in a substantial manner. Also, with larger values of $k$, the chance of this happening goes down. We train $\mathbf{E}$ to distinguish between $\mathbf{S}$ and $\mathbf{S}'$ by using a binary classifier on top of the representations generated by $\mathbf{E}$.

We extend the definition of consistency to pairs of sequences. Given two sequences, we define them to be consistent if they can be merged into a consistent sequence without changing the order of the tokens. Similar to the above definitions, we generate consistent and inconsistent pairs by starting from a consistent sequence and splitting them into two parts. We consider the following variations.

- ConsSent-N($k$): If $n$ is the number of tokens in a sequence $\mathbf{S}_1$, let $\mathbf{S}_1^1$ be a random subsequence of $\mathbf{S}_1$, and let $\mathbf{S}_1^2 = \mathbf{S}_1 \setminus \mathbf{S}_1^1$ be the complementary subsequence. For a consistent sequence $\mathbf{S}_1$, $\mathbf{S}_1^1$ and $\mathbf{S}_1^2$ form a consistent pairs of sequences. Let $\mathbf{S}_2^1$ and $\mathbf{S}_2^2$ be a partition for a different consistent sequence $\mathbf{S}_2$, such that $\mathbf{S}_1^2 \neq \mathbf{S}_2^2$. Then $\mathbf{S}_1^1$ and $\mathbf{S}_2^2$ form an inconsistent pair of sequences, by virtue of the fact they belong to two different consistent sequences. We can vary the complexity of the encoder $\mathbf{E}$ by training it to discriminate between a consistent pair $(\mathbf{S}_1^1, \mathbf{S}_1^2)$ and $k-1$ other inconsistent pairs $(\mathbf{S}_1^1, \mathbf{S}_2^2), (\mathbf{S}_1^1, \mathbf{S}_3^2), \cdots, (\mathbf{S}_1^1, \mathbf{S}_k^2)$ for different values of $k$.

  It is possible to pose the task of discriminating the consistent pair $(\mathbf{S}_1^1, \mathbf{S}_1^2)$ from the $k-1$ inconsistent pairs as a classification problem with a classification layer applied to encodings of the pairs. But this introduces additional parameters which is avoidable for sentence pairs. Instead, we train $\mathbf{E}$ by enforcing the constraint that

  $$\mathbf{E}(\mathbf{S}_1^1) \cdot \mathbf{E}(\mathbf{S}_1^2) \geq \mathbf{E}(\mathbf{S}_1^1) \cdot \mathbf{E}(\mathbf{S}_j^2) \quad \forall j \in \{2, k\}$$

  In other words, we train the encoder to place the representations of consistent pairs of sequences closer in terms of dot product than inconsistent pairs. A similar procedure was also used in training sentence representations in (Logeswaran & Lee (2018)), but with whole sentences appearing adjacent to each other in a larger body of text. Our method does not require the individual sentences to be ordered for training.

| Model | Positive Example | Negative Example |
|---|---|---|
| ConsSent-D(1) | Maya goes to school . | Maya goes to . |
| ConsSent-P(2) | Maya goes to school . | Maya to goes school . |
| ConsSent-I(1) | Maya goes to school . | Maya goes are to school. |
| ConsSent-R(1) | Maya goes to school . | Maya doesn't to school . |
| ConsSent-C(2) | Maya goes to school . | Maya it . |
|  | She loves it . | She loves goes to school . |
| ConsSent-N(2) | Maya goes to school . | Maya it school . |
|  | She loves it . | She loves goes to . |

Table 1: Example training sequences for the ConsSent encoders.

- ConsSent-C($k$): We generate $\mathbf{S}_1^1$ and $\mathbf{S}_1^2$ from $\mathbf{S}_1$ by partitioning it at a random point. Thus both $\mathbf{S}_1^1$ and $\mathbf{S}_1^2$ are contiguous subsequences of $\mathbf{S}_1$. If $\mathbf{S}_1$ is consistent, then the two partitioned sequences form a consistent pair. We generate inconsistent pairs by pairing $\mathbf{S}_1^1$ with $k - 1$ other $\mathbf{S}_j^2$ originating from the partition of different consistent sequences $\mathbf{S}_j$.

In addition to training an encoder on each individual task, we also consider training encoders in a multitask setting. We split the tasks into two groups, the first containing ConsSent-D($k$), ConsSent-P($k$), ConsSent-I($k$) and ConsSent-R($k$) and the other containing ConsSent-N($k$) and ConsSent-C($k$). We then train an encoder $\mathbf{E}_1$ for the first group and another encoder $\mathbf{E}_2$ for the second group, cycling through the tasks in a round robin manner within each group. Note that for the first group, we use different classification layers for the four different tasks. After training, the sentence representations from $\mathbf{E}_1$ and $\mathbf{E}_2$ is concatenated to generate the final representation for any sentence. We denote this model with multitask training by ConsSent-MT($k$). In Table1, we show toy positive and negative training examples for each of these training tasks. The choice of the encoder $\mathbf{E}$ is important to generate good sentence representations. Following Conneau et al. (2017), we use a bidirectional LSTM to process a sequence of tokens and take a max-pool of the intermediate hidden states to compute a distributed representation.

## 4 TRAINING AND EVALUATION

We train our models on the Billionword corpus (Chelba et al. (2014)). We use the first 50 shards of the training set (approximately 15 million sentences) for training and 50000 sentences from the validation set for validation. For ConsSent-D($k$), ConsSent-P($k$), ConsSent-I($k$) and ConsSent-R($k$) we delete, permute, insert or replace $k$ tokens with a probability of 0.5. This produces roughly an equal number of consistent and inconsistent sequences. For ConsSent-{D,I,K}($k$) we sweep over $k \in \{1, 2, 3, 4, 5\}$ and for ConsSent-P($k$) we sweep over $k \in \{2, 3, 4, 5, 6\}$ to train a total of 20 encoders.

In the case of ConsSent-N($k$), for each consistent sequence $\mathbf{S}_1$, we partition it into two subsequences by randomly picking a token to be in the first part $\mathbf{S}_1^1$ with probability 0.5. The remaining tokens go into $\mathbf{S}_1^2$. We pick $(k - 1)$ other random consistent sequences $\mathbf{S}_2, \cdots \mathbf{S}_k$ and do the same. For ConsSent-C($k$), for each consistent sequence $\mathbf{S}_1$, we pick $i \in \{2, \cdots, n - 1\}$ uniformly at random and partition it at $i$ to produce $\mathbf{S}_1^1$ and $\mathbf{S}_1^2$. The remainder of the training procedure is the same as ConsSent-N($k$). For both these models, we sweep over $k \in \{2, 3, 4, 5, 6\}$ to train a total of 10 encoders. For ConsSent-MT($k$), we sweep over $k \in \{2, 3, 4, 5, 6\}$. Overall, we train 35 encoders, 30 on the six individual tasks and 5 in the multitask setting. As described in the next section, we also consider an ensemble of models trained on different individual tasks.

We train a BiLSTM-Max encoder $\mathbf{E}$ with a hidden dimension of 2048 for each of the individual tasks, resulting in 4096 dimensional sentence representations. For ConsSent-MT($k$), we use a hidden dimension of 1024 for both $\mathbf{E}_1$ and $\mathbf{E}_2$, resulting in a sentence representation of 4096 dimensions. For ConsSent-D($k$), ConsSent-P($k$), ConsSent-I($k$) and ConsSent-R($k$), the sentence representations are passed through two linear layers of size 512 before the classification Softmax. For ConsSent-N($k$) and ConsSent-C($k$), we pair $\mathbf{S}_1^1$ with $(k - 1)$ random $\mathbf{S}_j^2$ from within the same minibatch to generate the inconsistent pairs. For optimization, we use SGD with an initial learning rate of 0.1

which is decayed by 0.99 after every epoch or by 0.2 if there is a drop in the validation accuracy. Gradients are clipped to a maximum norm of 5.0 and we train for a maximum of 20 epochs.

We evaluate the sentence encodings using the SentEval benchmark (Conneau & Kiela (2018)). This benchmark consists of two sets of tasks related to transfer learning and predicting linguistic properties of sentences. In the first set, there are 6 text classification tasks (MR, CR, SUBJ, MPQA, SST, TREC), one task on paraphrase detection (MRPC) and one on entailment classification (SICK-E). All these 8 tasks have accuracy as their performance measure. There are two other tasks on estimating the semantic relatedness of two sentences (SICK-R and STSB) for which the performance measure is Pearson correlation (expressed as percentage) between the estimated similarity scores and ground truth scores. For each of these datasets, the learned ConsSent sequence encoders are used to produce representations of sentences. These representations are then used for classification or score estimation using a logistic regression layer. We also use a L2 regularizer on the weights of the logistic layer whose coefficient is tuned using the validation sets. The goal of testing ConsSent on these tasks is to evaluate the quality of the sentence encoders on a wide variety of downstream tasks with limited training data.

The second set of tasks probes for 10 different linguistic properties of sentences. These include tasks like predicting which of a set of target words appears in a sentence (WordContent), the number of the subject in the main clause i.e. whether the subject is singular or plural (SubjNum), depth of the syntactic tree (TreeDepth) and length of the sentence quantized into a few bins (SentLen). Some of these properties are syntactic in nature, while some require deeper understanding of the semantics of a sentence. The goal of testing ConsSent on these tasks is to evaluate how much linguistic information is captured by the encoders. For each of the tasks, the representations produced by the ConsSent encoders are input to a classifier with a linear layer followed by Sigmoid followed by a classification layer. We tune the classifier on the validation sets by varying the dimension of the linear layer in $[50, 100, 200]$ and the dropout before the classification layer in $[0, 0.1, 0.2]$. For more details on these tasks, please refer to Conneau & Kiela (2018) and Conneau et al. (2018).

## 5    RESULTS ON TRANSFER TASKS

In Table 2 we present results on each of the transfer tasks in SentEval. In addition to the accuracy and correlation scores, we also report an average of all the 10 scores in the last column. We only show the best performing model (out of 5) for each of the six methods, based on the average validation performance over all the tasks in SentEval. ConsSent-N(3), which is trained to discriminate between pairs of sequences, performs the best on an average for any single task. ConsSent-MT(3) achieves a similar average performance, also achieving the best scores on five of the 10 transfer tasks among our models. Among the methods that classify single sequences, ConsSent-R(2) performs the best, second only to ConsSent-N(3). ConsSent-C is dominated by ConsSent-N in most cases, while ConsSent-D and ConsSent-P perform the worst.

Notably, all the methods perform better than SkipThought-LN (Kiros et al. (2015)) on an average and on most individual tasks. We also compare our results with QuickThoughts Logeswaran & Lee (2018). ConsSent-MT(3) outperforms QuickThoughts on MPQA (+0.2) and TREC(+0.8) and ConsSent-N(3) on MRPC(+0.8), while the latter performs better in the other tasks. It is important to note the differences between our models and QuickThoughts. Our approach only requires a set of unordered sentences, while QuickThoughts requires ordered sentences and this is crucial for its training. Further, our results were obtained by training on about 15M sentences, while QuickThoughts was trained on a much larger corpus of 181M sentences. We use a final sentence representation of 4096 dimensions while QuickThoughts uses 4800 dimensions.

As a baseline, we also train a LSTM (with 4096 hidden dimensions and 512 word embedding dimensions) language model on the same dataset that was used to train our models. The last hidden state of the trained LSTM acting on a sentence is then used as the sentence representation. As shown in Table 2, its performance is generally much worse than any of our models. This is not surprising as sentence encoders based on language models tend to work better when trained on large spans of text encompassing several ordered sentences, as has been observed in Radford (2018). The Billionword dataset lacks this property.

| Model | MR | CR | SUBJ | MPQA | SST | TREC | MRPC | SK-E | SK-R | STSB | AVG |
|---|---|---|---|---|---|---|---|---|---|---|---|
| *Unsupervised Methods* | | | | | | | | | | | |
| LangMod (Ours) | 72.1 | 72.0 | 87.8 | 88.1 | 77.4 | 75.0 | 75.4 | 77.7 | 70.3 | 54.4 | 75.0 |
| Untrained LSTM | 77.1 | 79.3 | 91.2 | 89.1 | 81.8 | 82.8 | 71.6 | 85.3 | 82.0 | 71.0 | 81.1 |
| FastText BoW | 78.2 | 80.2 | 91.8 | 88.0 | 82.3 | 83.4 | 74.4 | 82.0 | 78.9 | 70.2 | 80.9 |
| SkipThought-LN | 79.4 | 83.1 | 93.7 | 89.3 | 82.9 | 88.4 | 72.4 | 85.8 | 79.5 | 72.1 | 82.7 |
| QuickThoughts | 82.4 | 86.0 | 94.8 | 90.2 | 87.6 | 92.4 | 76.9 | - | 87.4 | - | - |
| *Our Methods* | | | | | | | | | | | |
| ConsSent-C(4) | 80.1 | 83.7 | 93.6 | 89.5 | 83.1 | 90.0 | 75.9 | **86.0** | 83.2 | 74.4 | 84.0 |
| ConsSent-N(3) | 80.1 | 84.2 | 93.8 | 89.5 | **83.4** | 90.8 | **77.3** | 86.1 | 83.8 | **75.8** | **84.4** |
| ConsSent-D(5) | 79.8 | 83.9 | 93.3 | 90.1 | 82.5 | 91.2 | 74.6 | 83.2 | 83.4 | 66.1 | 82.8 |
| ConsSent-P(3) | 80.0 | 83.2 | 93.4 | 89.9 | 82.8 | 92.2 | 75.4 | 84.0 | 83.1 | 68.6 | 83.3 |
| ConsSent-I(3) | **80.4** | 83.4 | 93.4 | 90.1 | 83.0 | 92.2 | 75.5 | 83.4 | 85.0 | 70.9 | 83.7 |
| ConsSent-R(2) | 79.9 | **84.3** | 93.5 | 90.2 | **83.4** | 92.6 | 75.9 | 84.2 | 85.2 | 72.0 | 84.1 |
| ConsSent-MT(3) | 80.2 | **84.3** | 94.4 | 90.4 | 83.1 | 93.1 | 76.7 | 83.4 | **86.5** | 72.2 | **84.4** |
| Ensemble | 81.6 | 85.1 | 94.4 | 90.6 | 85.2 | 93.8 | 77.7 | 86.8 | 87.2 | 77.3 | 86.0 |
| *Supervised Methods* | | | | | | | | | | | |
| InferSent | 81.1 | 86.3 | 92.4 | 90.2 | 84.6 | 88.2 | 76.2 | 86.3 | 88.4 | 75.8 | 85.0 |
| MultiTask | 82.5 | 87.7 | 94.0 | 90.9 | 83.2 | 93.0 | 78.6 | 87.8 | 88.8 | 78.9 | 86.5 |

Table 2: Performance of ConsSent on the transfer tasks in the SentEval benchmark. SkipThought is described in (Kiros et al., 2015), QuickThoughts in (Logeswaran & Lee, 2018) and MultiTask in Subramanian et al. (2018) and InferSent in Conneau et al. (2017). The other numbers (except LangMod) have been taken from Conneau et al. (2018). SK-R and SK-E stand for SICK-R and SICK-E respectively. AVG is a simple average over all the tasks. Bold indicates best result among our non-ensemble models and underline indicates best overall for unsupervised methods.

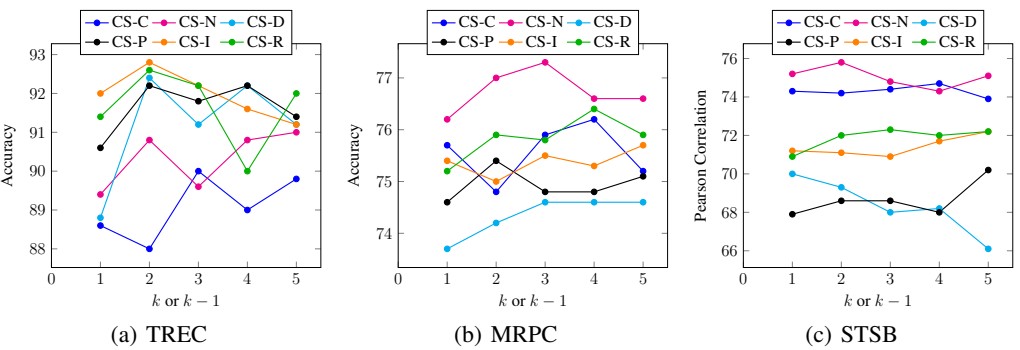

Figure 1: Performance of ConsSent models on the TREC, MRPC and STSB tasks. The y-values are in percentages. The x-values are $k$ for ConSent-{D,I,R}($k$) and $k-1$ for ConSent-{C,N,P}($k$).

Our best results are achieved by an ensemble of the six individual models in Table 2. The predictions of the ensemble model are calculated by weighting the predictions of the individual models by the normalized validation performance on each of the tasks in SentEval. The ensemble model shows improvements in all the tasks, with an average improvement of almost +1.6 over the best single model ConsSent-N(3) (or ConsSent-MT(3)). It also performs better than the supervised InferSent model and comparably to QuickThoughts. Interestingly, choosing the ensemble by using the best six models (out of the set of 30 models) based on validation performance gave slightly worse results. This empirically shows that the individual training tasks bias the sentence encoders in slightly different ways, and an ensemble of models can benefit from all of them.

We compare the relative performance of the encoders with varying values of $k$ for three of the transfer tasks - TREC, MRPC and STSB in Fig. 1. For TREC and MRPC, which are classification

| Model | SLen | WC | TDepth | TConst | BShift | Tense | SNum | ONum | SOMO | CInv | AVG |
|---|---|---|---|---|---|---|---|---|---|---|---|
| *Unsupervised Methods* | | | | | | | | | | | |
| LangMod (Ours) | 64.2 | 34.7 | 31.3 | 51.1 | 64.6 | 87.0 | 75.2 | 74.1 | 54.0 | 61.4 | 59.8 |
| Untrained | 73.3 | 88.8 | 46.2 | 71.8 | 70.6 | 89.2 | 85.8 | 81.9 | 73.3 | 68.3 | 74.9 |
| BoV-fastText | 66.6 | 91.6 | 37.1 | 68.1 | 50.8 | 89.1 | 82.1 | 79.8 | 54.2 | 54.8 | 67.4 |
| AutoEncoder | 99.4 | 16.8 | 46.3 | 75.2 | 71.9 | 87.7 | 88.5 | 86.5 | 73.5 | 72.4 | 71.8 |
| SkipThought | 79.1 | 48.4 | 45.7 | 79.2 | 73.4 | 90.7 | 86.6 | 81.7 | 72.4 | 72.3 | 73.0 |
| QuickThoughts | 90.6 | 90.3 | 40.2 | 80.7 | 56.8 | 86.2 | 83.0 | 79.7 | 55.3 | 70.0 | 73.3 |
| *Our Methods* | | | | | | | | | | | |
| ConsSent-C(4) | 82.6 | 87.2 | 44.6 | 80.9 | 67.3 | 89.2 | 87.7 | 80.4 | **63.6** | 71.0 | 75.4 |
| ConsSent-N(3) | 87.0 | 85.9 | 46.6 | 82.2 | 79.5 | 88.6 | 88.9 | 84.7 | 63.2 | 71.5 | 77.8 |
| ConsSent-D(5) | **89.5** | 88.2 | **54.9** | **84.9** | 84.5 | 90.0 | **90.0** | **88.5** | 61.5 | **73.2** | **80.5** |
| ConsSent-P(3) | 86.9 | 91.5 | 53.2 | 84.3 | 87.6 | **90.1** | 89.4 | 88.4 | 61.7 | 72.3 | **80.5** |
| ConsSent-I(3) | 85.7 | 92.3 | 52.4 | 83.5 | 85.6 | 89.7 | 88.8 | 87.2 | 62.1 | 71.5 | 79.9 |
| ConsSent-R(2) | 86.9 | 92.0 | 51.3 | 82.8 | 83.8 | **90.1** | 89.0 | 87.6 | 61.9 | 71.2 | 79.7 |
| ConsSent-MT(3) | 86.8 | **93.2** | 50.6 | 84.2 | **88.7** | 89.9 | 89.0 | 86.6 | 62.4 | 71.5 | 80.3 |
| Ensemble | 92.4 | 96.1 | 57.8 | 87.0 | 88.3 | 91.0 | 91.5 | 89.7 | 64.7 | 74.8 | 83.3 |
| *Supervised Methods* | | | | | | | | | | | |
| Seq2Tree | 96.5 | 8.7 | 62.0 | 88.9 | 83.6 | 91.5 | 94.5 | 94.3 | 73.5 | 73.8 | 76.7 |
| NMTEn-Fr | 80.1 | 58.3 | 51.7 | 81.9 | 73.7 | 89.5 | 90.3 | 89.1 | 73.2 | 75.4 | 76.3 |
| NLI | 71.7 | 87.3 | 41.6 | 70.5 | 65.1 | 86.7 | 80.7 | 80.3 | 62.1 | 66.8 | 71.3 |

Table 3: Performance of ConsSent on the linguistic probing tasks in the SentEval benchmark. Scores for QuickThoughts were obtained using the model provided by the authors. The other numbers (except LangMod) have been taken from (Conneau et al. (2018)).

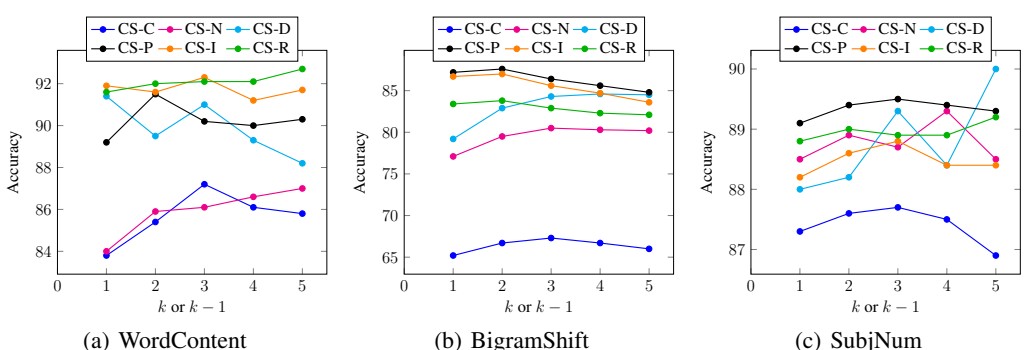

(a) WordContent      (b) BigramShift      (c) SubjNum

Figure 2: Performance of ConsSent models on the WordContent, BigramShift and SubjNum tasks. The y-values are in percentages. The x-values are $k$ for ConSent-$\{$D,I,R$\}(k)$ and $k-1$ for ConSent-$\{$C,N,P$\}(k)$.

tasks, there is roughly an inverted V shaped trend with some intermediate value of $k$ giving the best results for ConSent-D,P,I,R. Note that for smaller values of $k$, the encoders are exposed to negative examples that are are relatively similar to the positive ones and hence the discriminative training can be more noisy. On the other hand, as $k$ increases, the encoders may latch onto superficial patterns and hence not generalize well. For ConSent-C,N the trends are less clear for TREC but are closer to an inverted V for MRPC. For the semantic scoring task of STSB, the trend lines show no clear pattern.

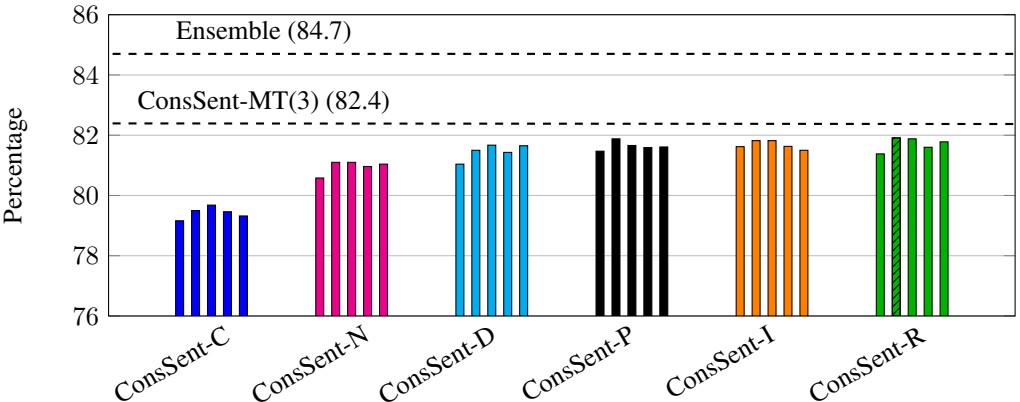

Figure 3: Average performance of our models over all the 20 tasks in SentEval. The bars in a group represent increasing values of $k$. The best performing model trained on a single task is ConsSent-R(2) (green with the black stripes) with an average score of 81.9.

## 6 RESULTS ON LINGUISTIC PROBING TASKS

We present results on the 10 linguistic probing tasks in SentEval in Table 3. All the encoders perform surprisingly well on most of the tasks, with the best ones ConsSent-D(5) and ConsSent-P(3) attaining an average score of 80.5. ConsSent-D(5) achieves the best score in five tasks among our models. ConsSent-MT(3) performs slightly worse, with an average score of 80.3. In general, encoders trained on single sequences perform better than the ones trained using pairs of sequences on the linguistic probing tasks.

Our methods perform significantly better than the other unsupervised methods including SkipThought and QuickThoughts. ConsSent-MT(3) achieves almost +7.0 points more average score than QuickThoughts, a model which does particularly well in the transfer tasks. Its performance is significantly better than even the supervised baseline results trained on machine translation (+4.0) and natural language entailment (+9.0) in Conneau et al. (2018). The performance of a third method Seq2Tree using a gated convolutional network (GCN) is however significantly better than the ConsSent encoders (except on the WordContent task). We have not experimented with a GCN encoder and it is possible that such an encoder may give better results in our case too. The sentence encoder trained on the language modeling objective performs poorly, with an average score of less than 60.

Here again, our best results are obtained by the ensemble of six models, which shows significant improvements over the individual models across most of the linguistic probing tasks, achieving an average score of 83.3, almost +2.8 more than the best single model ConsSent-D(5). In Fig. 2, we show the performance trends observed with varying values of $k$ for the encoders trained on individual tasks on three of the linguistic probing tasks – WordContent, BigramShift and SubjNum.

## 7 COMPARISON ACROSS TRANSFER AND LINGUISTIC PROBING TASKS

In this section, we compare the ConsSent models across the all the tasks in SentEval. In Fig.3 we plot the average performance of the encoders trained on the individual tasks, the multitask model and the ensemble model. Clearly, the sentence encoders trained on single sentences (ConsSent-D,P,I,R) perform better overall than those trained on pairs of sentences (ConsSent-C,N). In the former group, the ConsSent-R encoders tend to perform slightly better than the others, with ConsSent-R(2) achieving the best overall score of 81.9. The benefits of combining multiple tasks is borne out by the fact that ConsSent-MT(3) achieves the best score of 82.4 for a single encoder. The gains are even higher with the ensemble model, which achieves 84.7 or +2.8 points higher than the best model trained on a single task. Each of ConsSent-R(2), ConsSent-MT(3) and the ensemble model achieve significantly better scores on an average than existing unsupervised methods, including QuickThoughts, SkipThought and the baseline encoder trained with a language modeling objective on our dataset.

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
