# OpenReview forum: "Unsupervised Learning  of Sentence Representations Using Sequence Consistency"
_ICLR.cc/2019/Conference_

### Official Review · AnonReviewer1 · 2018-11-02
**The paper presents an unsupervised sentence encoding method trained to classify consistent (original) and inconsistent (corrupted) sentences. The trained encoders are used in a variety of tasks with good performance.**

**Rating:** 5
**Confidence:** 4

**Review:**

The paper presents an unsupervised sentence encoding method based on automatically generating inconsistent sentences by applying various transformations either to a single sentence or a pair and then training a model to classify the original sentences from the transformed ones.

Overall, I like the paper as it presents a simple method for training unsupervised sentence models which then can be used as part of further NLP tasks.

A few comments on the method and results:

- The results on Table 2 shows that supervised methods outperform unsupervised methods as well as the consistency based models with MultiTask having the largest margin. It would've been interesting to experiment with training multi-task layers on top of the sentence encoder and see how it would've performed.
- The detail of the architecture is slightly missing in a sense that it's not directly clear from the text if the output of the BiLSTMs is the final sentence encoding or the final layer before softmax?
- Also I would've thought that the output of LSTMs passed through nonlinear dense layers but the text refers to two linear layers.
- When I first read the paper, my eyes were looking for the result when you combine all of the transformations and train a single model :) - any reason why you didn't try this experiment?
- The paper is missing comparison and reference to recent works on universal language models (e.g. Radford et al 2018, Peters et al 2018, Howard et al 2018) as they rely on more elaborate model architectures and training compared to this paper but ultimately you can use them as sentence encoders.
- One final note, which could be a subsequent paper is to treat these transformations as part of an adversarial setup to further increase the robustness of a language model such as those mentioned previously.

---

> ### Author Response · Authors · 2018-11-23
> **Re: The trained encoders are used in a variety of tasks with good performance.**
>
> We thank the reviewer for reading the paper in detail and suggesting several improvements.
>
> 1. Multitask Training
>
> We have conducted further experiments by training models in a multitask setting. We split the six training tasks into two groups - one containing ConsSent-R(3), ConsSent-P(3), ConsSent-I(3), ConsSent-D(3) and the other containing ConsSent-N(3) and ConSent-C(3). We train two separate BiLSTM-Max encoders, one for each group, in a multitask setting by cycling through the tasks in a round-robin manner. Note that we use different classification layers for each of the tasks in the first group, keeping the sentence encoder LSTM same. The representations from the trained encoders are then concatenated to produce the final sentence representation for testing on SentEval. We use a hidden dimension of 1024 for each BiLSTM, thereby producing 4096 dimensional final sentence representations.
>
> The performance of the multitask trained model on the transfer tasks is the following.
>
> MR   CR     SUBJ MPQA SST  TREC MRPC SK-E SK-R  STSB AVG
> 80.2  84.3  94.4  90.4     83.1  93.1  76.7    83.4   86.5  72.2   84.4
>
> Compared to the best scores obtained by a single model in each task (see Table 2 in paper), the multitask model gains in some of the tasks e.g. SUBJ (+0.6), TREC (+0.3), SK-R (+1.3) but also suffers a drop in performance in some e.g. SST (-0.7), SK-E (-2.8) and STSB (-3.6). However, on a average, it matches the performance of the best single model ConsSent-N(3) with an average score of 84.4.
>
> The scores obtained for the linguistic probing tasks are as follows.
>
> SentLen WC   TDepth TConst BShift Tense SNum ONum SOMO CInv  AVG
> 86.8        93.2   50.6      84.2      88.7     89.9    89.0    86.6      62.4     71.5  80.3
>
> Here too, the multitask model gains in some cases e.g. WC (+1.1), BShift (+1.1) and suffers in some others e.g. SentLen (-2.7) and CInv (-1.7) over the best score by any single model. The average performance is slightly worse than the best single model on the linguistic probing tasks ConsSent-D(5) which achieves a score of 80.5.
>
> However, when we consider the average score over all the 20 tasks, the multitask model achieves a better score of 82.4 over the best single model ConsSent-R(2), gaining almost +0.5 points. This shows empirically that a model trained on a combination of tasks proposed in our paper can take advantage of the inductive biases of each of the tasks in a combined model.
>
> 2. The output of the sentence encoder is the output of the BiLSTMs. We will clarify this in the paper.
>
> 3. Our goal in training the sentence encoders in the unsupervised tasks is to allow them to learn general sentence representations that can be useful in a wide variety of other tasks. Using nonlinear activations in the classification layers while pertaining unduly biases the model towards the pertaining task, hurting its generalization ability. Such a scheme was also used in training InferSent (Conneau et al. 2017).
>
> References
> Conneau, Alexis et al. “Supervised Learning of Universal Sentence Representations from Natural Language Inference Data.” EMNLP (2017).

---

> > ### Author Response · Authors · 2018-11-23
> > **Re: The trained encoders are used in a variety of tasks with good performance.**
> >
> > continued from above....
> >
> > 4. As mentioned in point (1) above, we have conducted further experiments in a multitask setting, thereby producing a single model with better average performance than any one of the single models. We have also evaluated the performance of an ensemble of the single models. We pick the six best models, one from each training task, based on the average validation performance on all the 20 tasks in SentEval. These are ConsSent-R(2), ConsSent-P(3), ConsSent-I(3), ConsSent-D(5), ConsSent-N(3) and ConSent-C(4). We then create an ensemble model by weighting the predicted probabilities from each model by the normalized validation performance on each of the tasks in SentEval. The performance of the ensemble model on the transfer tasks is the following
> >
> > MR       CR      SUBJ    MPQA     SST     TREC   MRPC   SK-E   SK-R    STSB   AVG
> > 81.6     85.1    94.4     90.6        85.2     93.8     77.7      86.8    87.2    77.3     86.0
> > (+1.0)  (+0.8)  (+0.6)  (+0.3)     (+1.4)  (+1.0)  (+0.4)   (+0.6)  (+2.0)  (+1.5)  (+1.6)  (<-- improvement over best score by any single model. See Table 2 in paper.)
> >
> > The ensemble model shows improvements in all the tasks, with an average improvement of almost +1.6 points over the best single model ConsSent-N(3). The improvements for MR (+1.0), SST (+1.4), TREC (+1.0), SK-R (+2.0) and STSB (+1.5) over the best score obtained by a single model are particularly significant.
> >
> > The performance of the ensemble model on the linguistic probing tasks is the following
> >
> > SentLen WC     TDepth  TConst  BShift  Tense  SNum   ONum   SOMO   CInv    AVG
> > 92.4       96.1     57.8       87.0       88.3      91.0     91.5     89.7        64.7        74.8     83.3
> > (+2.9)    (+4.0)   (+2.9)    (+2.1)    (+0.7)   (+0.9)   (+1.5)   (+1.2) .   (+1.1)    (+1.6)  (+2.8)   (<-- improvement over best score by any single model. See Table 2 in paper.)
> >
> > The ensemble model shows strong improvements across all the tasks, with an average improvement of +2.8 over the best single model ConsSent-D(5). The improvements for SentLen(+2.9), WC(+4.0), TDepth(+2.9) and TConst(+2.1) over the best score obtained by a single model are particularly significant.
> >
> > The average performance of the ensemble over all the 20 tasks is 84.7, which is an improvement of +2.8 over the best single model ConsSent-R(2). Interestingly, choosing the ensemble by using the best six models (out of the set of 30 models) based on validation performance gave slightly worse results. This empirically shows that the individual training tasks bias the sentence encoders in slightly different ways, and an ensemble of models can benefit from all of them.
> >
> > We will add these results in the final version of the paper, to be updated shortly.

---

### Official Review · AnonReviewer3 · 2018-11-02
**Interesting idea to learn sentence representations**

**Rating:** 5
**Confidence:** 4

**Review:**

This submission presents a model for self-supervised learning of sentence representations. The core idea is to train a sentence encoder to predict sequence consistency. Sentences from a text corpus are considered consistent (positive examples), while simple editions of these make the negative samples. Six different ways to edit the sequence are proposed. The network is trained to solve this binary classification task, separately for all six possible editions.
The proposed approach is evaluated on SentEval giving encouraging results.

+ The proposed approach is interesting. It is similar in some sense to the self-supervised representation learning literature in computer vision, where the network is trained to say- predict the rotation applied to the image.

- If one considers that sentence encoders can be trained using a pretext task, this paper lacks a very-simple-yet-hard-to-beat baseline. Unlike for images, natural language has a very natural self-supervised task: language modeling. Results reported for language-modeling-based sentence representations outperform results reported in the tables by a big margin. Here is at least one paper that would be worth mentioning:
- Radford, Alec, Rafal Jozefowicz, and Ilya Sutskever. "Learning to generate reviews and discovering sentiment." arXiv preprint arXiv:1704.01444 (2017).
In order to make things comparable, it would be good to provide reference numbers for an LSTM trained with a LM objective on the same data as the experiments in this paper.

- If I understood correctly, all variants are trained separately (for each of the 6 different ways to edit the sequence). This makes the reading of the results very hard. Table 2 should not contain all possible variants, but one single solution that works best according to some criterion.
To this end, why would these models be trained separately? First of all, the main result could be an ensemble of all 6, or the model could be made multi-class, or even multi-label, capable of predicting all variants in a single task.

Overall, I think that this paper proposes an interesting alternative for training sentence representations. However, the execution of the paper lacks in several respects outlines above. Therefore, I lean towards rejection, and await the other reviews, comments and answer from the authors to make my final decision.

---

> ### Author Response · Authors · 2018-11-23
> **Re: Interesting idea to learn sentence representations**
>
> We thank the reviewer for reading the paper carefully and providing with constructive suggestions.
>
> Comparison with Language Modeling based sentence encoders
>
> We agree that language modeling is a natural unsupervised objective for natural language and large pertained language models can be used as effective sentence encoders. One key advantage of the objectives we propose is that they can be learned using corpora of unordered sentences. For language modeling based sentence encoders to work, it is important to train them on large continuous spans of text straddling several sentences, as mentioned in (Radford et al. 2018). In fact, when we train a baseline LSTM language model (single layer with 4096 hidden dimensions) on the same dataset we used, and evaluated the sentence representations on SentEval, the results (shown below) were significantly worse than any of the models proposed in our paper.
>
> MR   CR    SUBJ  MPQA SST   TREC  MRPC  SK-E  SK-R  STSB  AVG
> 72.1 72.0  87.8   88.1     77.4  75.0   75.4      77.7  70.3 .  54.4  75.0
>
> SentLen WC   TDepth TConst BShift Tense SNum ONum SOMO CInv  AVG
> 64.2         34.7 31.3       51.1      64.6     87.0    75.2    74.1      54.0    61.4   59.8
>
> Alternately, it will be worthwhile to train our models on larger spans of text across multiple sentences from BookCorpus and evaluate them. This will be part of our future work.
>
> Combining models and training tasks
>
> We have conducted further experiments on combining the different models and tasks. We do so in two different ways.
>
> 1. Ensemble of the six best models
>
> We pick the six best models, one from each training task, based on the average validation performance on all the 20 tasks in SentEval. These are ConsSent-R(2), ConsSent-P(3), ConsSent-I(3), ConsSent-D(5), ConsSent-N(3) and ConSent-C(4). We then create an ensemble model by weighting the predicted probabilities from each model by the normalized validation performance on each of the tasks in SentEval. The performance of the ensemble model on the transfer tasks is the following
>
> MR       CR      SUBJ    MPQA     SST     TREC   MRPC   SK-E   SK-R    STSB   AVG
> 81.6     85.1    94.4     90.6        85.2     93.8     77.7      86.8    87.2    77.3     86.0
> (+1.0)  (+0.8)  (+0.6)  (+0.3)     (+1.4)  (+1.0)  (+0.4)   (+0.6)  (+2.0)  (+1.5)  (+1.6)  (<-- improvement over best score by any single model. See Table 2 in paper.)
>
> The ensemble model shows improvements in all the tasks, with an average improvement of almost +1.6 points over the best single model ConsSent-N(3). The improvements for MR (+1.0), SST (+1.4), TREC (+1.0), SK-R (+2.0) and STSB (+1.5) over the best score obtained by a single model are particularly significant.
>
> The performance of the ensemble model on the linguistic probing tasks is the following
>
> SentLen WC     TDepth  TConst  BShift  Tense  SNum   ONum   SOMO   CInv    AVG
> 92.4       96.1     57.8       87.0       88.3      91.0     91.5     89.7        64.7        74.8     83.3
> (+2.9)    (+4.0)   (+2.9)    (+2.1)    (+0.7)   (+0.9)   (+1.5)   (+1.2) .   (+1.1)    (+1.6)  (+2.8)   (<-- improvement over best score by any single model. See Table 2 in paper.)
>
> The ensemble model shows strong improvements across all the tasks, with an average improvement of +2.8 over the best single model ConsSent-D(5). The improvements for SentLen(+2.9), WC(+4.0), TDepth(+2.9) and TConst(+2.1) over the best score obtained by a single model are particularly significant.
>
> The average performance of the ensemble over all the 20 tasks is 84.7, which is an improvement of +2.8 over the best single model ConsSent-R(2). Interestingly, choosing the ensemble by using the best six models (out of the set of 30 models) based on validation performance gave slightly worse results. This empirically shows that the individual training tasks bias the sentence encoders in slightly different ways, and an ensemble of models can benefit from all of them.

---

> > ### Author Response · Authors · 2018-11-23
> > **Re: Interesting idea to learn sentence representations**
> >
> > Continuing from above...
> >
> > 2. MultiTask training
> >
> > Multitask learning has been shown to give strong results for sentence representation learning in the supervised setting (Subramanian et al. 2018). We split the six training tasks into two groups - one containing ConsSent-R(3), ConsSent-P(3), ConsSent-I(3), ConsSent-D(3) and the other containing ConsSent-N(3) and ConSent-C(3). We train two separate BiLSTM-Max encoders, one for each group, in a multitask setting by cycling through the tasks in a round-robin manner. Note that we use different classification layers for each of the tasks in the first group, keeping the sentence encoder LSTM same. The representations from the trained encoders are then concatenated to produce the final sentence representation for testing on SentEval. We use a hidden dimension of 1024 for each BiLSTM, thereby producing 4096 dimensional final sentence representations.
> >
> > The performance of the multitask trained model on the transfer tasks is the following.
> >
> > MR   CR    SUBJ MPQA SST  TREC MRPC SK-E SK-R STSB AVG
> > 80.2  84.3 94.4  90.4     83.1  93.1  76.7    83.4   86.5 72.2   84.4
> >
> > Compared to the best scores obtained by a single model in each task (see Table 2 in paper), the multitask model gains in some of the tasks e.g. SUBJ (+0.6), TREC (+0.3), SK-R (+1.3) but also suffers a drop in performance in some e.g. SST (-0.7), SK-E (-2.8) and STSB (-3.6). However, on a average, it matches the performance of the best single model ConsSent-N(3) with an average score of 84.4.
> >
> > The scores obtained for the linguistic probing tasks are as follows.
> >
> > SentLen WC   TDepth TConst BShift Tense SNum ONum SOMO CInv  AVG
> > 86.8        93.2   50.6      84.2      88.7     89.9    89.0    86.6     62.4      71.5  80.3
> >
> > Here too, the multitask model gains in some cases e.g. WC (+1.1), BShift (+1.1) and suffers in some others e.g. SentLen (-2.7) and CInv (-1.7) over the best score by any single model. The average performance is slightly worse than the best single model on the linguistic probing tasks ConsSent-D(5) which achieves a score of 80.5.
> >
> > However, when we consider the average score over all the 20 tasks, the multitask model achieves a better score of 82.4 over the best single model ConsSent-R(2), gaining almost +0.5 points. This shows empirically that a model trained on a combination of tasks proposed in our paper can take advantage of the inductive biases of each of the tasks in a combined model.
> >
> > To summarize, both an ensemble of models trained on the six different tasks and a single model trained in a multitask setting show improvements over any single model, the former more strongly so. We will add these results in the updated version of the paper.

---

### Official Review · AnonReviewer2 · 2018-11-02
**Simple method for learning sentence representations, with competitive results**

**Rating:** 7
**Confidence:** 4

**Review:**


== Clarity ==
The primary strength of this paper is the simplicity of the approach.

Main idea #1: corrupt sentences (via random insertions/deletions/permutations), and train a sentence encoder to determine whether a sentence has been corrupted or not.

Main idea #2: split a sentence into two parts (two different ways to do this were proposed). Train a sequence encoder to encode each part such that we can tell whether the two parts came from the same sentence or not.

I can see that this would be very easy for others to implement, perhaps encouraging its adoption.

== Quality of results ==
The proposed approach is evaluated on the well-known SentEval benchmark.

It generally does not outperform supervised approaches such as InferSent and MultiTask. However, this is fine because the proposed approach uses no supervised data, and can be applied in domains/languages where supervised data is not available.

The approach is competitive with existing state-of-the-art sentence representations such as QuickThoughts. However, it is not definitively better:

Out of the 9 tasks with results for QuickThoughts, this approach (ConsSent) performs better on 3 (MPQA +0.1%, TREC +0.4%, MRPC +0.4%). For the other 6 tasks, ConsSent performs worse (MR -1.8%, CR -1.7%, SUBJ -1%, SST -3.8%, SK-R, -2.4%). Taken together, the losses seem to be larger than the gains.

Furthermore, the QuickThoughts results were obtained with a single model across all SentEval tasks. In contrast, the ConsSent approach requires a different hyperparameter setting for each task in order to achieve comparable results -- there is no single hyperparameter setting that would give state-of-the-art results across all tasks.

The authors also evaluate on the newly-released linguistic probing tasks in SentEval. They strongly outperform several existing methods on this benchmark. However, it is unclear why they did not compare against QuickThoughts, which was the strongest baseline on the original SentEval tasks.

== Originality ==
The proposed approach is simple and straightforward. This is on the whole a great thing, but perhaps not especially surprising from an originality/novelty perspective.

Therefore, the significance and impact of this approach really needs to be carried by the quality of the empirical results.

The sentence pair based approaches (ConsSent-N and C) are conceptually interesting, but don't seem to be responsible for the best results on the linguistic probing tasks.

== Conclusion ==

Pros:
- conceptual simplicity
- competitive results (better than many previous unsup. sentence representation methods, excluding QuickThoughts)
- strong results on SentEval's linguistic probing task

Cons:
- no single hyperparameter value (perturbation method and value for k) gets great results across all tasks
- some important baselines possibly missing for linguistic probing tasks

---

> ### Author Response · Authors · 2018-11-23
> **Re: Simple method for learning sentence representations, with competitive results**
>
> We thank the reviewer for a careful reading of our paper and the detailed comments. We have conducted further experiments on combining the different models and tasks. We do so in two different ways.
>
> 1. Ensemble of six models, one from each task
>
> We pick the six best models, one from each training task, based on the average validation performance on all the 20 tasks in SentEval. These are ConsSent-R(2), ConsSent-P(3), ConsSent-I(3), ConsSent-D(5), ConsSent-N(3) and ConSent-C(4). We then create an ensemble model by weighting the predicted probabilities from each model by the normalized validation performance on each of the tasks in SentEval. The performance of the ensemble model on the transfer tasks is the following
>
> MR       CR      SUBJ    MPQA     SST     TREC   MRPC   SK-E   SK-R    STSB   AVG
> 81.6     85.1    94.4     90.6        85.2     93.8     77.7      86.8    87.2    77.3     86.0
> (+1.0)  (+0.8)  (+0.6)  (+0.3)     (+1.4)  (+1.0)  (+0.4)   (+0.6)  (+2.0)  (+1.5)  (+1.6)  (<-- improvement over best score by any single model. See Table 2 in paper.)
>
> The ensemble model shows improvements in all the tasks, with an average improvement of almost +1.6 points over the best single model ConsSent-N(3). The improvements for MR (+1.0), SST (+1.4), TREC (+1.0), SK-R (+2.0) and STSB (+1.5) over the best score obtained by a single model are particularly significant.
>
> The performance of the ensemble model on the linguistic probing tasks is the following
>
> SentLen WC     TDepth  TConst  BShift  Tense  SNum   ONum   SOMO   CInv    AVG
> 92.4       96.1     57.8       87.0       88.3      91.0     91.5     89.7        64.7        74.8     83.3
> (+2.9)    (+4.0)   (+2.9)    (+2.1)    (+0.7)   (+0.9)   (+1.5)   (+1.2) .   (+1.1)    (+1.6)  (+2.8)   (<-- improvement over best score by any single model. See Table 2 in paper.)
>
> The ensemble model shows strong improvements across all the tasks, with an average improvement of +2.8 over the best single model ConsSent-D(5). The improvements for SentLen(+2.9), WC(+4.0), TDepth(+2.9) and TConst(+2.1) over the best score obtained by a single model are particularly significant.
>
> The average performance of the ensemble over all the 20 tasks is 84.7, which is an improvement of +2.8 over the best single model ConsSent-R(2). Interestingly, choosing the ensemble by using the best six models (out of the set of 30 models) based on validation performance gave slightly worse results. This empirically shows that the individual training tasks bias the sentence encoders in slightly different ways, and an ensemble of models can benefit from all of them.

---

> > ### Author Response · Authors · 2018-11-23
> > **Re: Simple method for learning sentence representations, with competitive results**
> >
> > Continuing from previous comment...
> >
> > 2. MultiTask learning
> >
> > Multitask learning has been shown to give strong results for sentence representation learning in the supervised setting (Subramanian et al. 2018). We split the six training tasks into two groups - one containing ConsSent-R(3), ConsSent-P(3), ConsSent-I(3), ConsSent-D(3) and the other containing ConsSent-N(3) and ConSent-C(3). We train two separate BiLSTM-Max encoders, one for each group, in a multitask setting by cycling through the tasks in a round-robin manner. Note that we use different classification layers for each of the tasks in the first group, keeping the sentence encoder LSTM same. The representations from the trained encoders are then concatenated to produce the final sentence representation for testing on SentEval. We use a hidden dimension of 1024 for each BiLSTM, thereby producing 4096 dimensional final sentence representations.
> >
> > The performance of the multitask trained model on the transfer tasks is the following.
> >
> > MR   CR   SUBJ  MPQA SST  TREC MRPC SK-E  SK-R  STSB AVG
> > 80.2  84.3 94.4  90.4     83.1  93.1  76.7    83.4   86.5  72.2   84.4
> >
> > Compared to the best scores obtained by a single model in each task (see Table 2 in paper), the multitask model gains in some of the tasks e.g. SUBJ (+0.6), TREC (+0.3), SK-R (+1.3) but also suffers a drop in performance in some e.g. SST (-0.7), SK-E (-2.8) and STSB (-3.6). However, on a average, it matches the performance of the best single model ConsSent-N(3) with an average score of 84.4.
> >
> >
> > The scores obtained for the linguistic probing tasks are as follows.
> >
> > SentLen WC   TDepth TConst BShift Tense SNum ONum SOMO CInv  AVG
> > 86.8        93.2   50.6      84.2      88.7     89.9    89.0    86.6     62.4     71.5    80.3
> >
> > Here too, the multitask model gains in some cases e.g. WC (+1.1), BShift (+1.1) and suffers in some others e.g. SentLen (-2.7) and CInv (-1.7) over the best score by any single model. The average performance is slightly worse than the best single model on the linguistic probing tasks ConsSent-D(5) which achieves a score of 80.5.
> >
> > However, when we consider the average score over all the 20 tasks, the multitask model achieves a better score of 82.4 over the best single model ConsSent-R(2), gaining almost +0.5 points. This shows empirically that a model trained on a combination of tasks proposed in our paper can take advantage of the inductive biases of each of the tasks in a combined model.
> >
> > To summarize, both an ensemble of models trained on the six different tasks and a single model trained in a multitask setting show improvements over any single model, the former more strongly so.
> >
> >
> > 3. Comparison with QuickThoughts
> >
> > There are some differences between our approach and QuickThoughts. Our approach only requires a set of unordered sentences, while QuickThoughts requires ordered sentences and this is crucial for its training. Further, our results were obtained by training on about 15M sentences, while QuickThoughts was trained using much larger corpora - 45M sentences in BookCorpus and 126M in UMBC. We use a final sentence representation of 4096 dimensions while QuickThoughts uses 4800 dimensions. It will be worthwhile to train our models on the BookCorpus+UMBC dataset and compare performance, which will be part of our future work.
> >
> > As suggested by the reviewer, we also took the trained models made available by the authors of QuickThoughts (at https://github.com/lajanugen/S2V) and computed its performance on the linguistic probing tasks. We followed the same protocol that was used to evaluate our models.
> >
> > SentLen  WC     TDepth  TConst BShift  Tense  SNum   ONum  SOMO   CInv   AVG
> > 90.6        90.3     40.2        80.7      56.8      86.2     83.0      79.7      55.3       70.0     73.3
> > (+1.1)    (+2.1)   (-14.7)    (-4.2)    (-27.7)  (-3.8)   (-7.0)     (-8.8)   (-6.2)      (-3.2)  (-7.2). (<-- QuickThoughts as compared to ConsSent-D(5))
> >
> > Somewhat surprisingly, the  average performance was about 73.3, comparable to SkipThought (73.0), but significantly worse than ConsSent-D(5) which has an average score of 80.5. Thus, although QuickThoughts achieves better performance on the transfer tasks, the sentence representations learned by it captures much less linguistic information.
> >
> > We will add these results in the updated version of the paper, which will be posted shortly.
> >
> > References
> >
> > Subramanian, S., Trischler, A., Bengio, Y., & Pal, C.J. (2018). Learning General Purpose Distributed Sentence Representations via Large Scale Multi-task Learning. CoRR, abs/1804.00079.

---

### Author Response · Authors · 2018-11-25
**Revised version uploaded**

We have uploaded a revised and improved version of the paper, incorporating most of the suggestions made by the reviewers. We have already given detailed comments below, but to recap, the revised version contains the following changes

1. Results for a multitask trained encoder, which gives the best average performance over all the tasks in SentEval for a single model, have been added.

2. Results for an ensemble model comprising of six encoders trained on individual tasks have been added. These represent the best results in the paper, comparable to QuickThoughts in the transfer tasks and much better in the linguistic probing tasks.

3. Baseline results for an encoder trained on a language model objective on our dataset have been added. The performance of the encoder is poor because the dataset lacks long contexts of ordered sentences. In contrast, our methods can be trained with unordered sentences.

4. Additional related work has been added in Section 2, especially on the language model based encoders.

5. Section 7 has been condensed to include only one figure showing the average performance of our models over all the 20 tasks in SentEval.

6. Several typos have also been corrected.

---

### Meta-Review · Area_Chair1 · 2018-12-14
**Not quite enough for acceptance**

**Confidence:** 5
**Recommendation:** Reject

**Metareview:**

The overall view of the reviewers is that the paper is not quite good enough as it stands. The reviewers also appreciates the contributions so taking the comments into account and resubmit elsewhere is encouraged.